# Lithocholic Acid Induces miR21, Promoting PTEN Inhibition via STAT3 and ERK-1/2 Signaling in Colorectal Cancer Cells

**DOI:** 10.3390/ijms221910209

**Published:** 2021-09-22

**Authors:** Thinh-Thi Nguyen, Thuan-Trong Ung, Shinan Li, Dhiraj Kumar Sah, Sun-Young Park, Sen Lian, Young-Do Jung

**Affiliations:** 1Research Institute of Medical Sciences, Chonnam National University Medical School, Gwangju 501-190, Korea; thinhnt1984@gmail.com (T.-T.N.); ungtrongthuan@gmail.com (T.-T.U.); shinanli@foxmail.com (S.L.); dhiraj007sah@gmail.com (D.K.S.); drpsy@naver.com (S.-Y.P.); 2Nanogen Pharmaceutical Biotechnology Joint Stock Company, Ho Chi Minh City 71207, Vietnam; 3Department of Biochemistry and Molecular Biology, School of Basic Medical Sciences, Southern Medical University, Guangzhou 510515, China

**Keywords:** miR21, lithocholic acid, colorectal cancer, STAT3, PTEN

## Abstract

Micro-RNA-21 (miR-21) is a vital regulator of colorectal cancer (CRC) progression and has emerged as a potential therapeutic target in CRC treatment. Our study using real-time PCR assay found that a secondary bile acid, lithocholic acid (LCA), stimulated the expression of miR21 in the CRC cell lines. Promoter activity assay showed that LCA strongly stimulated miR21 promoter activity in HCT116 cells in a time- and dose-dependent manner. Studies of chemical inhibitors and miR21 promoter mutants indicated that Erk1/2 signaling, AP-1 transcription factor, and STAT3 are major signals involved in the mechanism of LCA-induced miR21 in HCT116 cells. The elevation of miR21 expression was upstream of the phosphatase and tensin homolog (PTEN) inhibition, and CRC cell proliferation enhancement that was shown to be possibly mediated by PI3K/AKT signaling activation. This study is the first to report that LCA affects miR21 expression in CRC cells, providing us with a better understanding of the cancer-promoting mechanism of bile acids that have been described as the very first promoters of CRC progression.

## 1. Introduction

Micro-RNA-21 (miR-21) is an abundantly expressed microRNA across various mammalian cells [1,2] and is one of the earliest identified cancer-promoting “oncomiRs”. The carcinogenesis mechanism of miR-21 has been documented to target numerous tumor suppressor genes, including *PTEN*, *PDCD4*, *TIMP3*, and *RHOB*, which are associated with cell proliferation, migration, invasion, metastasis, and apoptosis [3], or are known to play important roles in signaling pathways such as the RAS/MEK/ERK, PTEN/PI3K/AKT, and Wnt/β-catenin pathways [4]. 

In a large-scale profiling of miRNA expression in 540 human samples derived from 363 specimens representing six types of solid tumors and 177 respective normal control tissues, miR-21 was the only miRNA upregulated in all analyzed tumors types, including colorectal carcinoma [2]. More recent studies have indicated that miR-21 is also upregulated in leukemic cancers [5].

In CRC, high levels of tumoral miR-21 expression are associated with poor prognosis as well as poor response to chemotherapy. Moreover, high levels of serum miR-21 distinguish patients with adenomas and CRCs from healthy control subjects. Interestingly, a higher concentration of miR-21 was found in the blood drawn near the site of primary tumor compared to that in the peripheral blood, indicating that the primary tumor releases a high concentration of miR-21, which is diluted in the circulatory system [6]. 

Accumulating evidence shows that miR-21 is a vital regulator in the development of CRC through regulation of the proliferation, invasion, migration, and apoptosis of CRC cells, and it has emerged as a novel potential therapeutic target for CRC treatment. Mima et al. found that the miR-21 expression level in CRC is associated with poor clinical outcomes, and this association is stronger in carcinomas expressing high levels of miR21, which play complex roles in immunity and inflammation in tumor progression [7]. Furthermore, Li et al. reported that miR-21 is overexpressed in CRC cell lines and promotes the proliferation, migration, and invasion of these cells in vitro, which is associated with downregulation of Sec23A expression involved in anti-tumorigenesis [4]. Dig et al. observed that antisense oligonucleotides against miR-21 effectively inhibit the growth and metastasis of CRC in vivo and this is accompanied by a downregulated expression of miR-21 and reduced transduction of the AKT and ERK pathways. Mechanically, global gene expression analysis showed that the expression of DUSP8, a novel target of miR-21, is upregulated in tumor mass [8].

Bile acids (BAs) are amphiphilic derivatives of cholesterol and have been associated with increased incidence of CRC. Though there are no studies on CRC cells, studies on liver and pancreatic cells revealed that the miRNA expression profile is significantly modified under the effect of BAs [9,10]. In this study, we revealed that LCA-induced the expression of miR21 via STAT3 inhibition and Erk1/2 activation. In turn, the upregulated miR21 inhibited tumor suppressor protein expression, PTEN, and stimulated the proliferation of CRC cells that was possibly mediated via AKT/PI3K activation.

## 2. Results

### 2.1. LCA Induces miR21 Expression in CRC Cell Lines

To investigate the effect of BAs on miR21 expression, HCT116 cells were treated with 20 µM LCA, deoxycholic acid (DCA), cholic acid (CA), or chenodeoxycholic acid (CDCA) within 24 h of incubation in DMEM media containing 1% FBS. HCT116 cells were treated with 100 nM of PMA as a positive control. The treated cells were then harvested, total RNA was extracted, and miR21 expression was checked using Real-time PCR. The results show that secondary BAs significantly induce the expression of miR21 in HCT116 cells, and LCA shows a stronger stimulatory effect on miR21 expression compared to that of DCA, even showing a competitive effect compared to that with PMA, a strong tumor promoter (Figure 1A,B). Therefore, we selected LCA for the subsequent studies on the effect of BAs on miR21 expression.

We consequently checked the effect of LCA on miR21 expression in different CRC cell lines including SW480, HCT15, DLD1, HT29, and HCT116. All these cells were treated with 30 µM LCA within 24 h and then the miR21 expression was determined. The Real-time PCR results show that LCA has the strongest stimulatory effect on miR21 expression in HCT116 cells (Appendix A). Therefore, we selected HCT116 cells as the cell model for further studies.

We performed further experiments to check the effect of LCA on miR21 at different concentrations and different time periods. The results show that LCA stimulates the expression of miR21 in HCT116 cells in a concentration- (Figure 1C,D) and time-dependent manner (Appendix A). The optimal condition to observe maximized stimulatory effect of LCA on miR21 expression was with 30 µM LCA within 24 h of incubation. 

### 2.2. AP-1 Transcription Factor Is Involved in LCA-Induced miR21 Promoter Activity

To determine whether LCA produced any effect on miR21 promoter activity, we constructed the miR21 promoter on pGL3 plasmid backbone as the promoter of luciferase reporter gene and assessed its activity under LCA treatment. LCA exhibited a strong stimulatory effect on miR21 promoter activity in a concentration- (Figure 2A) and time-dependent manner (Appendix A). Accordingly, incubating HCT116 cells with 30 µM LCA within 18 h resulted in the highest enhancement of miR21 promoter activity. 

As described in a previous study [11], the miR21 promoter includes three binding sites for the AP-1 transcription factor (Figure 2A). As such, we hypothesized that this factor could be involved in LCA-induced miR21 promoter activity. To confirm this hypothesis, we constructed miR21 promoters with three deletion mutants of each AP-1 binding site including ΔAP1-1, ΔAP1-2, and ΔAP1-3, and one deletion mutant of all three binding sites, ΔAP1-1&2&3 (Figure 2B). These reporter plasmids and the plasmid containing the original miR21 promoter were transfected into HCT116 cells to observe the stimulatory effect of LCA on their activities. The results show that the miR21 promoters with deletion mutants of the second and third AP-1 binding sites, ΔAP1-2 and ΔAP1-3, are less activated by the LCA treatment than that by the original miR21 promoter. Moreover, the miR21 promoter containing the first AP-1 binding site mutant did not show any stimulatory effect by LCA, as also observed for the triple AP1 binding site deletion mutant promoter, ΔAP1-1&2&3. These results suggest that AP-1 transcription factor plays a major role in the mechanism of LCA-induced miR21 expression, with the first binding site of AP-1 possibly playing the most important role.

To confirm the effect of the AP-1 transcription factor in LCA-induced miR21 expression, we observed the effect of LCA on miR21 expression using Real-time PCR in the presence of SR11302 as an AP-1 inhibitor. The result shows that SR11302 blocks the LCA stimulatory effect on miR21 expression (Figure 2C), suggesting that LCA-induced miR21 expression is associated with the AP-1 transcription factor.

### 2.3. Erk1/2 Signal Is Involved in LCA-Induced miR21 Expression through AP-1 Transcription Factor Activation

In a previous study, we reported that LCA stimulated Erk1/2 signaling, which is upstream of the AP-1 transcription factor activation [12]. Here, we tested whether the Erk1/2 signal was involved in LCA-induced miR21 expression. The experiment with an Erk1/2 inhibitor, PD98059, showed that it abrogated the LCA stimulatory effect on miR21 promoter activity (Figure 3C) as well as miR21 expression, assessed by Real-time PCR (Figure 3A,B). 

In addition, the dominant mutant of Erk1/2 (K97M) impaired the miR21 promoter activity stimulated by LCA (Appendix A). Moreover, Erk1/2 was also confirmed to be involved in the activation of the AP-1 transcription factor by LCA. The results show that the inhibitor (PD98059) and dominant mutant of Erk1/2 (K97M) significantly inhibit the AP-1 promoter activation triggered by LCA (Appendix A). These results indicate that LCA upregulates miR21 expression in HCT116 via Erk1/2/AP-1 signaling.

### 2.4. Role of STAT3 in LCA-Induced miR21

STAT3 signal was shown to be inhibited by LCA and its inhibition is mediated by the activation of Erk1/2 signaling in HCT116 cells [12]. There are two binding sites for STAT3 on the miR21 promoter, mapped near the Pri-miR21 transcription start point (Figure 4A). We hypothesized that STAT3 could therefore be involved in LCA-induced miR21 expression. To verify this hypothesis, we transfected siSTAT3 into HCT116 cells and then observed the miR21 expression under LCA treatment. The results show that the miR21 expression (Figure 4A,B) and miR21 promoter activity (Appendix A) in STAT3-blocked HCT116 cells are significantly enhanced compared to those in the original cells. Consistently, a STAT3-activator, interleukin-6, inhibited LCA-induced miR21 expression in a concentration-dependent manner (Appendix A). These results suggest that STAT3 signal is involved in LCA-induced miR21 expression in HCT116 cells. To achieve a better understanding of the role of STAT3 in this mechanism, we constructed miR21 promoters mutated at two binding sites of STAT3, miR21P-mutSTAT3-BS1 and miR21P-mutSTAT3-BS2 (Figure 4C). The promoter activity assays showed that the miR21P-mutSTAT3-BS1 promoter was more significantly stimulated by LCA than that by the original miR21P promoter and the miR21P-mutSTAT3-BS2 promoter. This result suggests that the interaction of phospho-STAT3 and miR21 promoter lead of a negative regulation of miR21 expression. The presence of the secondary BA, LCA, inhibited the STAT3 activation, preventing it from binding to the miR21 promoter, and finally releasing the transcription process of *PrimiR21* gene. 

Based on all obtained results from the study, we established the signaling mechanism of LCA on miR21 expression as follows: LCA inhibits STAT3 phosphorylation and prevents it from binding to the miR21 promoter. On the other hand, LCA stimulates Erk1/2 signaling for AP-1 transcription factor activation that could also mediate for STAT3 inhibition. Both the interactions of AP-1 transcription factor with miR21 promoter and the release of the STAT3 binding site from its partner, activated STAT3, cooperatively promote the transcription process of the *PrimiR21* gene, leading to the observed upregulation of mature miR21.

### 2.5. LCA Inhibits the PTEN Expression Due to LCA-Induced miR21 in HCT116 Cells

It is well known that PTEN is one of the downstream targets of miR21. Specifically, miR21 downregulates the expression of this protein. Therefore, we determined how LCA impacted PTEN expression in CRC cell lines. By RT-PCR assays, we found that LCA significantly inhibited PTEN expression in HCT116 cells (Figure 5C) and other CRC cell lines (Appendix A). Both Real-time PCR (Figure 5A,B) and Western blot analysis (Figure 5D) also consistently showed that the expression of PTEN was inhibited in the LCA-treated cells in a concentration-dependent manner.

We performed inhibitor experiments to investigate the signals involved in LCA-induced PTEN. As predicted, inhibitors of the Erk1/2 signal (PD98059) and the AP-1 transcription factor (SR11302) suppressed the effect of LCA on PTEN expression, as confirmed by both Real-time PCR results (Figure 6A,C) and Western blot results (Figure 6B,D and Appendix A). In addition, siSTAT3, a STAT3 silencer, synergistically supported the inhibitory effect of LCA on PTEN expression. Moreover, the basal PTEN expression in the STAT3-inactivated HCT116 cells was also lowered by siSTAT3 transfection compared to that in the original cells (Figure 6E). These results suggest that LCA downregulates PTEN expression via Erk1/2/AP-1 activation and STAT3 inhibition, similar to that by LCA on miR21 expression. 

In light of these results, we predicted that PTEN inhibition was downstream of LCA-induced miR21 expression. To confirm this, we blocked miR21 expression by transfecting miR21 inhibitor into HCT116 cells (Figure 7A,B). miR21 inhibitor was able to rescue PTEN expression from LCA treatment at both transcriptional (Figure 7C) and translational levels (Figure 7D). These results demonstrate that LCA upregulates miR21 expression and then causes the inhibitory effect on PTEN expression in CRC HCT116 cells.

### 2.6. LCA Enhances the Proliferation of CRC HCT116 Cells via miR21 Upregulation 

One of the downstream signals of miR21 is AKT/PI3K, proved to be one of the main players regulating cell proliferation. Therefore, we performed Western blot assays to observe the activation of AKT/PI3k signaling under LCA treatment. The result shows that LCA strongly activates AKT/PI3k signaling in a time-dependent manner (Figure 8A). Moreover, LCA also stimulated CRC HCT116 cell proliferation, and this effect was eradicated in the cells transfected with miR21 inhibitor (Figure 8B). These results suggest that LCA induced miR21 expression which, in turn, stimulated the cell proliferation enhancement of CRC HCT116 cells.

## 3. Discussion

miR21 is an “onco-miRNA” and an upregulated level of this miRNA is considered to be a biomarker for the progression of different cancer types [13,14]. BAs have been well documented to be cancer-promoting, with substantial studies having been conducted on their molecular mechanisms for identifying their molecular-leveled targets [15]. However, the impact of BAs on miRNA in general and miR21 in particular is still not fully known. Yuan et al. first reported that BAs moderate miR21 expression. The study showed that miR21 expression of gastric cancer cells increases when cells are treated with DCA, leading to SOX2 expression suppression and the simultaneous induction of CDX2 expression under BA treatment [16]. In agreement, Rodrigues et al. also observed that DCA, at a low dose (25 µM), produces no toxicity in the cells and slightly induces miR21 expression in primary rat hepatocytes, whereas moderate to high doses (>100 mM) significantly inhibited the miRNA expression [17]. Our group was the first to report the effect of BAs on miR21 expression in CRC cells. These results identified another target of BAs in their CRC oncogenesis mechanism, emphasizing their impacts as promoters of CRC and other cancers.

Not only is miR21 moderated by BAs, but it is also known to be regulated by driving factors of BA circulation, lipid accumulation [18,19,20] and gut microbiota dysbacteriosis [21,22]. In particular, Wu et al. proved that miR21 is a potential link between lipid accumulation and hepatocellular carcinoma. It is increased in the livers of high fat diet-fed mice and human HepG2 cells incubated with fatty acid. The increased expression of this miRNA promotes hepatic lipid accumulation and cancer progression by interacting with the Hbp1-p53-Srebp1c pathway [20]. The gut microbiota are known to utilize BAs and their conjugates resulting in a smaller, unconjugated hydrophobic BA pool which accumulates at a concentration of around 500 μM [23]. In addition, it is documented that the gut microbiota participate in the process of colorectal inflammation and CRC by regulating miRNAs [24]. Nakata et al. found that commensal bacteria increased miR-21-5p expression level and promoted intestinal epithelial permeability by regulating ARF4 [21]. In this study, we proved that LCA, a secondary BA, stimulates miR21 expression in CRC cells. This is an important proof that could further support to clear the picture of CRC development, starting from the gut microbiota involvement, in which BAs and miR21 could be two key mediators of the process.

Erk1/2 signaling and AP-1 transcription factor are well-known major signals involved in the mechanisms of LCA [12,25] and other BAs [15]. In this study, these signals were further shown to be involved in the LCA-induced miR21 mechanism. These results are consistent with previous studies that reported the involvement of the Erk1/2 pathway and AP-1 transcription factor in the miR21 expression-modulating mechanism stimulated by the *hepatitis C* virus, the S100P/RAGE signaling pathway, and dehydroepiandrosterone, a ligand for a G-protein-coupled receptor [26,27,28]. In a similar approach, Mercado-Pimentel et al. constructed mutants of the miR21 promoter at three binding sites of AP-1 transcription factors to assess the role of AP-1 in miR21 expression regulated by S100P/RAGE signaling in CRC. The study confirmed the involvement of AP-1 in the miR21-regulation mechanism, but it did not elucidate the different effects of the three AP-1 binding sites [27]. In this study, the first binding site of AP-1 was shown to play a decisive role in the upregulated miR21 expression stimulated by LCA. We also obtained consistent result of the experiment with DCA treatment. Contrary to the naive miR21 promoter construct, the miR21 promoter activation could not be observed under DCA treatment in the ΔAP1-1 mutant promoter construct. These results confirmed the involvement of AP-1 signaling in the BA-induced miR21 mechanism, in which the first binding site of AP-1 possibly plays a fundamental role. 

The transcription factor family of STAT proteins has a crucial regulatory role in numerous biological processes, such as cell proliferation, differentiation, and survival, and it is critical in malignant transformation and oncogenesis [29]. STAT3 has been found to be activated inappropriately in a wide range of human cancers, including CRC. Potential downstream STAT3 targets include genes that regulate cell cycle progression, cell survival/growth, and angiogenesis [30]. It was recently shown that STAT3 can also regulate the expression of the cancer-related miR-21 in multiple myeloma [31]. The upstream enhancer of the *PimiR21* gene was found to contain two STAT3 binding sites that are strictly conserved [11]. There are many studies that have shown that the interaction between active STAT3 and the miR21 promoter via their specific binding sites enhanced miR21 transcription, leading to the elevated level of this miRNA [32,33,34,35]. However, Ohno et al. showed an inverse relationship between STAT3 and miR21 expression. The authors observed that a STAT3-specific inhibitory peptide increased the level of miR-21 expression and inhibited IFN-β-mediated suppression of miR-21. The addition of the STAT3 silencer prior to IFN-β treatment rescued the activity of the miR21 promoter suppressed by IFN-β treatment. Moreover, the STAT3- expressing vector transfected into the cells significantly reduced the miR21 promoter activity, and the addition of IFN-β further suppressed it [36]. Another study by Li and Zeng also reported a negative correlation between miR-21 and the JAK/STAT signal pathway in systemic juvenile idiopathic arthritis (JIA). While the expression of miR-21 was significantly lower in JIA patients than that in healthy controls, the level of STAT3 increased in peripheral blood mononuclear cells of the JIA group compared with that in the control group [37]. In agreement with the two studies, our results show a negative regulation of the miR21 expression by STAT3. Mutated miR21 promoters at STAT3 binding sites, as well as STAT3 inactivation by siSTAT3 application enhanced LCA-induced miR21 expression, whereas IL-6 treatment tended to abrogate the stimulatory effect of LCA on miR21. These results are consistent with previous studies [12,38,39] that reported STAT3 as a negative regulator of intestinal tumor progression. 

With the extensive studies on the molecular biology of cancer, researchers have found that PTEN/PI3K/AKT signal pathways are closely associated with the growth, proliferation, infiltration, expansion, and metastasis of malignant cells [40]. Recent research shows that miR-21, via PTEN/PI3K/AKT signal pathways, influences the biological behaviors of malignant cells, especially cell proliferation [41,42,43,44]. The PI3K/AKT and MAPK pathways have been well documented to be major intracellular signaling pathways stimulated by BAs [45,46]. In CRC cells, PI3K/AKT signaling is activated by conjugated BA and deoxycholyltaurine and then stimulated in CRC cell survival and proliferation. Moreover, conjugated BAs also inhibit programmed cell death by multiple PI3K/AKT-mediated mechanisms including the phosphorylation of glycogen synthase kinase 3 and NF-κB activation [47]. In agreement with previous studies, this study also observed PI3K/AKT activation and CRC cell proliferation acceleration stimulated by LCA. miR21 upregulation was proven to be the key factor for the cell proliferation enhancement, but the role of PI3K/AKT signaling in this picture still remains unclear.

In this study, LCA was proved to upregulate miR21 expression via Erk1/2/AP-1 and STAT3 signaling in CRC HCT116 cells, in turn inhibiting PTEN expression and enhancing cell proliferation. This is a newly discovered signal targeted by BAs, thus determining the role of BAs as promoters of CRC progression. This finding provides us with a deeper understanding of LCA carcinogenicity and contributes to the development of new strategies for CRC treatment.

## 4. Materials and Methods

### 4.1. Cell Culture Conditions and Materials

Human colon carcinoma HCT116, DLD1, HCT15, HT29, and SW480 cell lines were obtained from the American Type Culture Collection (Rockville, MD, USA). The cells were cultured at 37 °C in a 5% CO_2_ atmosphere in McCoy’s 5A medium supplemented with 10% fetal bovine serum (FBS) and 1% penicillin-streptomycin. CA, CDCA, DCA, and LCA were obtained from Sigma Chemical Co. (Saint Louis, MO, USA) and dissolved in dimethyl sulfoxide (DMSO) as 30 mM stock solutions. PD98059 (Calbiochem, San Diego, CA, USA) and SR11302 (Bio-Techne, Minneapolis, MN, USA) were dissolved in DMSO as 30 mM stocks, and stored at −80 °C. SignalSilence^®^ Stat3 siRNA II, SignalSilence^®^ Control siRNA (Cell Signaling Technology, Danvers, MA, USA) and hsa-21 miRNA inhibitor, inhibitor negative control (Applied Biological Materials Inc., Richmond, BC, Canada) were re-suspended in DNase and RNase-free ddH_2_O to prepare 100 µM stock and stored at −20 °C. The dominant-negative mutant of Erk1/2 (K97M) was kindly provided by Dr. N. G. Ahn (University of Colorado Boulder, Boulder, CO, USA).

### 4.2. miR21 Expression Using Real-Time PCR

Total RNAs of CRC cells were extracted using TRIzol RNA Isolation Reagents (Thermo Fisher Scientific, Waltham, MA, USA). The miR21 level was determined using the HB miR21 assay kit™ system II (Heimbiotek, Seongnam-si, Gyeonggi-do, Korea), following the manufacturer’s instructions. The levels of small nuclear RNA (snRNA), RU6B, were determined using HB RU6B assay kit™ system II (Heimbiotek) as the internal control to normalize miR21 levels among different samples. We used 500 ng of total RNA for miR21-cDNA synthesis. Then, 1 µL of each cDNA product sample was used for Real-time PCR assay using Rotor-Gene Q machine (Qiagen, Hilden, Germany). Reactions were typically run in duplicate. The data were analyzed on Rotor-Gene Q Series software 2.3.1. The cycle number at which the reaction crossed an arbitrarily-placed threshold (*C*_T_) was determined for each gene and the relative amount of miR21 to RU6B was described using 2^−Δ*C*_T_^, where Δ*C*_T_ = (*C*_T-miR21_ − *C*_T-RU6B_).

### 4.3. PTEN Transcriptional Expression

PTEN mRNA level were checked using both Real-time PCR and RT-PCR. A sample of 1 µg of total RNA extracted by TRIzol RNA Isolation Reagents was used for first-strand complementary DNA synthesis using random primers and M-MLV transcriptase (Promega, Madison, WI, USA).

For the RT-PCR assay, the complementary DNA was subjected to PCR amplification with primer sets for PTEN and GAPDH as an internal control, using a PCR master mix solution (iNtRON Biotechnology, Seongnam-Si, Gyeonggi-do, Korea). The specific primers sequences for PTEN and GAPDH are listed in Appendix A.

In parallel, PTEN mRNA levels were quantified using Real-time PCR with 2× SYBR green PCR master mix (Qiagen) and the PTEN specific primers. To normalize PTEN expression among different RNA samples, GAPDH level was also determined as an internal control. The reactions were run in duplicate using Rotor-Gene Q machine (Qiagen). The cycle number at which the reaction crossed *C*_T_ was determined for each gene and the relative amount of PTEN to GAPDH was calculated using 2^−Δ*C*_T_^, where Δ*C*_T_ = (*C*_T-miR21_ − *C*_T-RU6B_).

### 4.4. miR21 Promoter Construction

We constructed the miR21 promoter from −503 to −1 before the transcription start position of *PrimiR21* gene using the primer pair miR21 Pro-503-F/miR21 Pro-R listed in Appendix A.

The 503bp fragment of the miR21 promoter was amplified from the genome of the HCT116 cell line through a PCR reaction with the above primers. The PCR product was then purified and cloned into a TA vector, pCR2.1^®^, (ThermoFisher Scientific, Carlsbad, CA, USA) to select the correct sequence based on sequencing results. The correct sequence of the 503bp-miR21 promoter was then inserted into a pGL3 plasmid using two restriction enzymes, KpnI and HindIII. The successful constructions of the miR21 promoter reporter plasmid (pGL3/miR21P) were confirmed using colony PCR reaction with the above primer pair and cutting reaction with the KpnI/HindIII restriction enzyme pair.

### 4.5. miR21 Promoter Deletion Mutant Construction

The sequences of the miR21 promoter with deletion mutants at different AP-1 binding sites including ΔAP1-1, ΔAP1-2, ΔAP1-3, and ΔAP1-1&2&3 were constructed by applying overlapping PCR method using pGL3/miR21P plasmid as the reaction template and primers listed in Appendix A. All obtained products from the PCR reaction were cloned into pCR2.1^®^ vector, sequenced to confirm the expected sequences, and then inserted into a pGL3 plasmid using KpnI/HindIII restriction enzymes, similar to the miR21 promoter construction procedure described above.

The binding sites of STAT3 on the miR21 promoter were site-mutated using the Phusion Site-Directed Mutagenesis Kit (ThermoFisher Scientific), pGL3/miR21P plasmid as the reaction template, and the primers listed in Appendix A. The PCR products were treated with DpnI enzyme to digest the original pGL3/miR21P plasmid and then transformed into *Escherichia coli* DH5α. Plasmids were extracted, purified, and sent for sequencing to screen those with expected mutations. The selected plasmids were treated with KpnI/HindIII to obtain miR21 promoter sequences including expected mutations, and then inserted into a pGL3 plasmid backbone using restriction enzymes.

### 4.6. Promoter Activity Assay

Transcriptional regulation of miR21 by LCA was examined by transient transfection of 250 ng pGL3/miR21P or its mutants, including pGL3/miR21P-ΔAP1-1, pGL3/miR21P-ΔAP1-2, pGL3/miR21P-ΔAP1-3, pGL3/miR21P-ΔAP1-1&2&3, pGL3/miR21P-mutSTAT3-BS1, and pGL3/miR21P -mutSTAT3-BS2. The promoter activities stimulated by LCA treatment were determined following procedures described in previous studies [12]. The effects of Erk1/2 inhibitor (PD98059) and AP-1 inhibitor (SR11302) on miR21 promoter activity were determined by pre-treating the cells with inhibitors an hour prior to the addition of LCA. In addition, 1 µg of Erk1/2 dominant mutant (K97M) was co-transfected with 250 ng of pGL3/miR21P into HCT116 cells to ascertain the role of Erk1/2 signaling in LCA-induced miR21 expression. pSV2neo, the expression plasmid vector of K97M mutant, was used as construct control for K97M mutant transfection.

### 4.7. Western Blot Analysis

Total proteins were extracted from HCT116 cells using PRO-PREP™ protein extraction solution (iNtRON Biotechnology). Then, 15–30 μg of protein was resolved on 12% sodium dodecyl sulfate-polyacrylamide gel and the expression levels of PTEN, phospho-AKT, and phospho-PI3K were determined using Western blot, as described in previous studies [48]. The applied primary antibodies included rabbit polyclonal anti-PTEN, anti-phospho AKT, and anti-phospho PI3K (Cell Signaling Technology). The effects of Erk1/2 inhibitor (PD98059), AP-1 inhibitor (SR11302), on PTEN expression were determined by pre-treatment of cells with the inhibitor for 1 h prior to the addition of LCA. miR21 inhibitor was transfected into HCT116 cells for 24 h before LCA treatment to the miR21 role on LCA- inhibited PTEN.

### 4.8. Transfection

hsa-21 miRNA inhibitor or SignalSilence^®^ Stat3 siRNA II was transiently transfected into HCT116 cells with lipofectamin 2000 transfection reagent (ThermoFisher Scientific), following the manufacturer’s instructions. Briefly, HCT116 cells were grown in DMEM containing 10% PBS for them to reach 30–50% confluency at the time of transfection. miR21 inhibitor or siSTAT3 were prepared with lipofectamin 2000. The final mixture had a final concentration of miR21 inhibitor or siSTAT3 of 50 nM and lipofectamin 2000 was diluted 100-fold. Then, 24 h post-transfection, the cells were subjected to LCA to evaluate its effect on miR21 expression or miR21 promoter activity, as described above. SignalSilence^®^ Control siRNA and inhibitor negative control were, respectively used as construct controls of siSTAT3, and miR21 inhibitor in the transfection experiments.

### 4.9. Proliferation Assay

HCT116 cells transfected with miR21 inhibitor or the inhibitor negative control were treated with 30 µM LCA for 24 h, and then the cell proliferation was assessed using the Cell Viability, Proliferation and Cytotoxicity Assay Kit (EZ-Cytox, DoGenBio, Seoul, Korea) based on the WST reagent. The EZ-Cytox reagent was diluted 10-fold in FBS free DMEM media and incubated with the cells. The plates were read in a microreader (ELISA) machine at 450 nm at different time points of 30 min, 1 h, 2 h, 4 h, and 8 h to check cell proliferation.

### 4.10. Densitometric Analysis

The visualized bands obtained after immunoblotting experiments were submitted to densitometric analysis using imageJ analysis software (National Institute of Health). β-actin was used for normalization of the immunoblotting products. Results are expressed as relative protein expression to β-actin.

### 4.11. Statistical Analysis

All results reflect a minimum of three independent experiments. We performed analysis of variance for multivariable analyses. *p* < 0.05 was considered statistically significant.

## Figures and Tables

**Figure 1 ijms-22-10209-f001:**
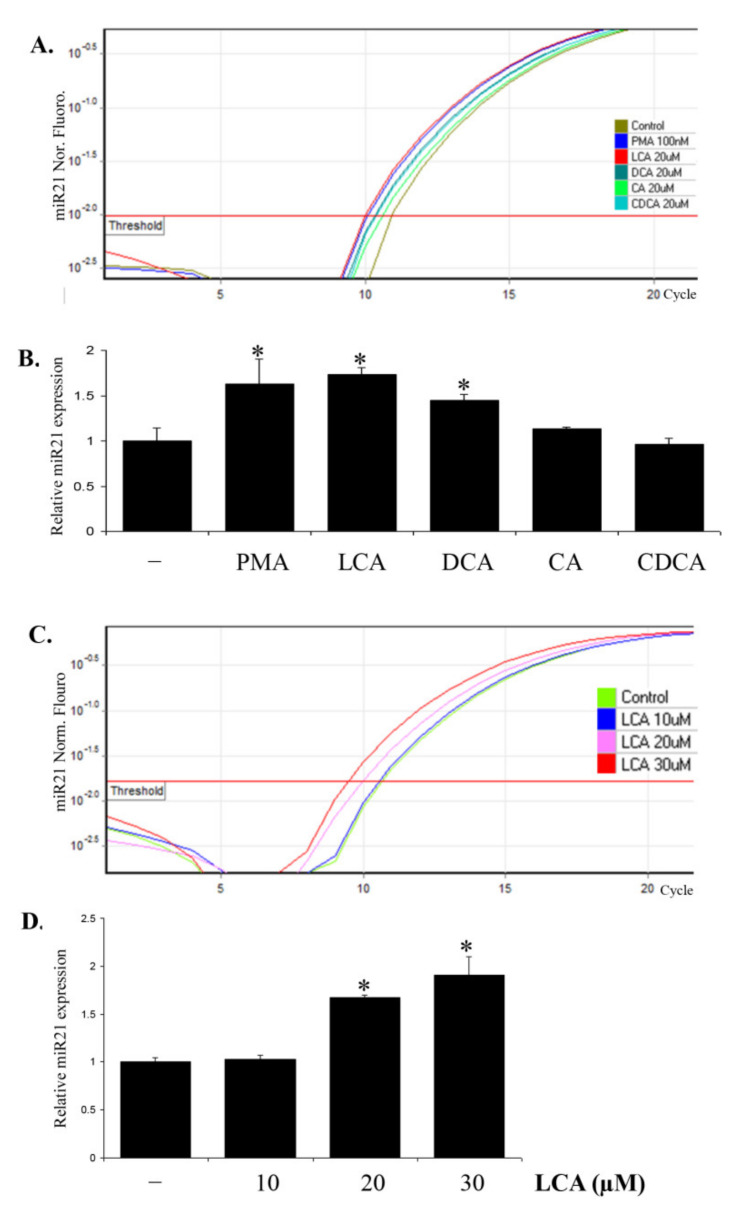
LCA induces miR21 expression in HCT116 cells. (**A**,**B**) HCT116 cells were incubated with indicated compounds including 100 nM PMA and 20 µM of different BAs for 24 h. The cells were then extracted for total RNA for checking miR21 expression by Realtime-PCR. (**C**,**D**) HCT116 cells were incubated with LCA at different concentrations for 24 h and submitted to Realtime-PCR for miR21 expression. * *p* < 0.05 versus control. The above data represent the means ± SD from triplicate measurements.

**Figure 2 ijms-22-10209-f002:**
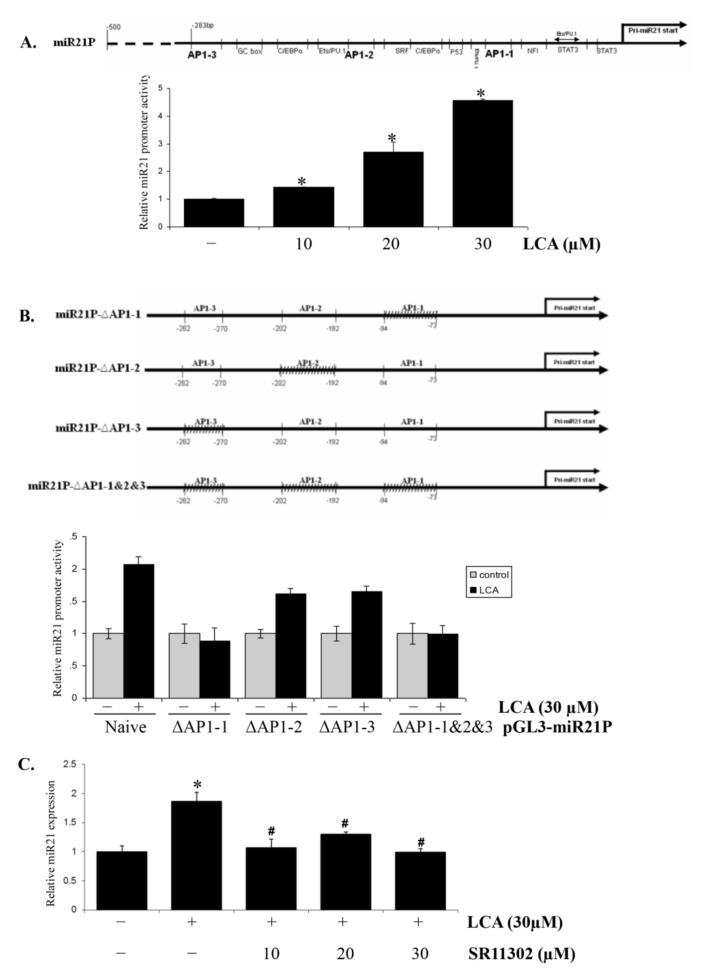
The involvement of AP-1 transcription factor in LCA-induced miR21 promoter activity. (**A**) HCT116 cells were transiently transfected with pGL3-miR21 promoter and then treated with 10 µM–30 µM LCA for 18 h. The cells were then lysed and checked for miR21 promoter activity by luciferase assay. (**B**) HCT116 cells were transiently transfected with pGL3-miR21 promoter and its mutants at AP-1 binding sites including ΔAP1-1, ΔAP1-2, ΔAP1-3, ΔAP1-1&2&3. The cells were then treated with 30 µM LCA for 18 h and submitted to dual-luciferase assay for checking miR21 promoter activity. (**C**) HCT116 cells were treated with AP-1 specific inhibitor, SR 11,302 1 h prior to 30 µM LCA treatment for 24 h. The cells were then extracted for total RNA and checked for miR21 expression by Realtime-PCR. * *p* < 0.05 versus control; # *p* < 0.05 versus LCA. The above data represent the means ± SD from triplicate measurements.

**Figure 3 ijms-22-10209-f003:**
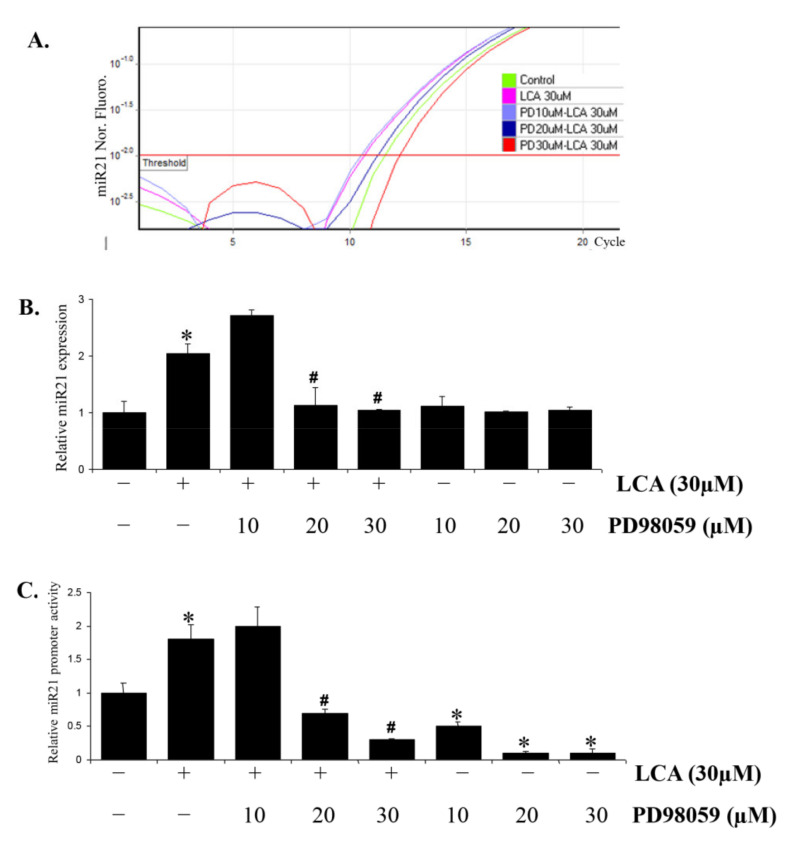
Erk1/2 signaling is involved in LCA-induced miR21 through AP-1 transcription factor activation. (**A**,**B**) HCT116 cells were treated to PD98059 1 h prior to LCA treatment for 24 h. The cells were then harvested and checked for miR21 expression by Realtime-PCR. (**C**) HCT116 cells transfected with pGL3-miR21 promoter were treated with PD98059 and then 30 µM LCA for 24 h. The cells were then lysed and subjected for dual-luciferase assay for miR21 promoter activity. * *p* < 0.05 versus control; # *p* < 0.05 versus only LCA. The above data represent the means ± SD from triplicate measurements.

**Figure 4 ijms-22-10209-f004:**
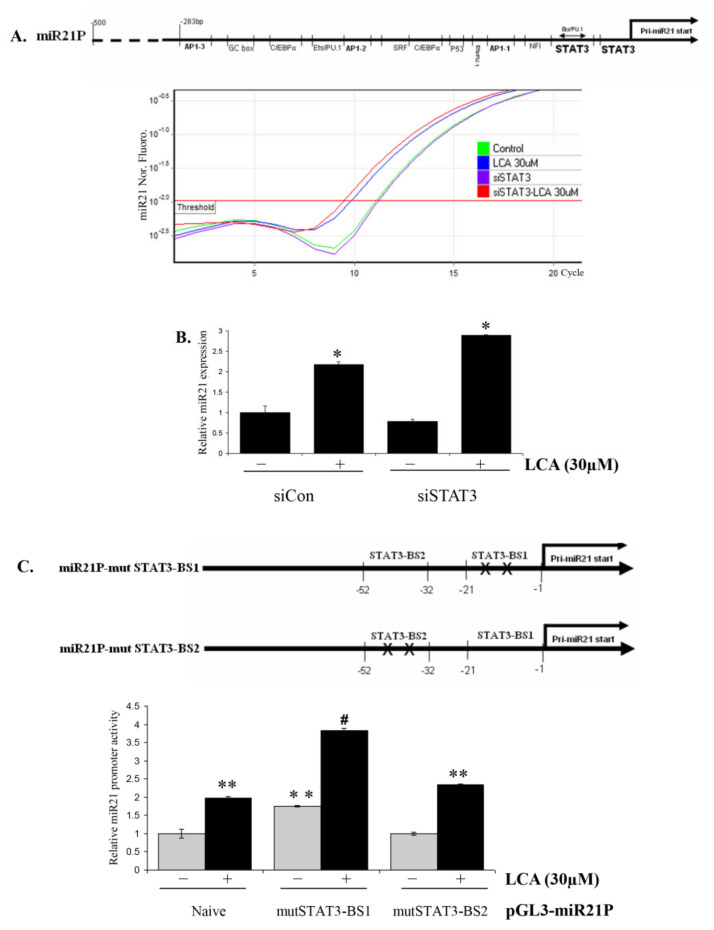
STAT3 signaling is involved in LCA-induced miR21 expression in HCT116 cells. (**A**,**B**) HCT116 cells were transiently transfected with 50 nM control siRNA (siCon) or siSTAT3 by Lipofactamin 2000 for 24 h. The cells were then treated with 30 µM LCA for 24 h and checked for miR21 expression by Realtime-PCR. * *p* < 0.05 versus control (**C**) HCT116 cells were transiently transfected with pGL3-miR21 promoter and its mutants at STAT3 binding sites including mutSTAT3-BS1, mutSTAT3-BS2. The cells were then treated with 30 µM LCA for 18 h and checked for miR21 promoter activity. ** *p* < 0.05 versus control of cells transfected with pGEMT/miR21P-naive plasmid, # *p* < 0.05 versus LCA of cells transfected with pGEMT/miR21P-naïve plasmid. The above data represent the means ± SD from triplicate measurements.

**Figure 5 ijms-22-10209-f005:**
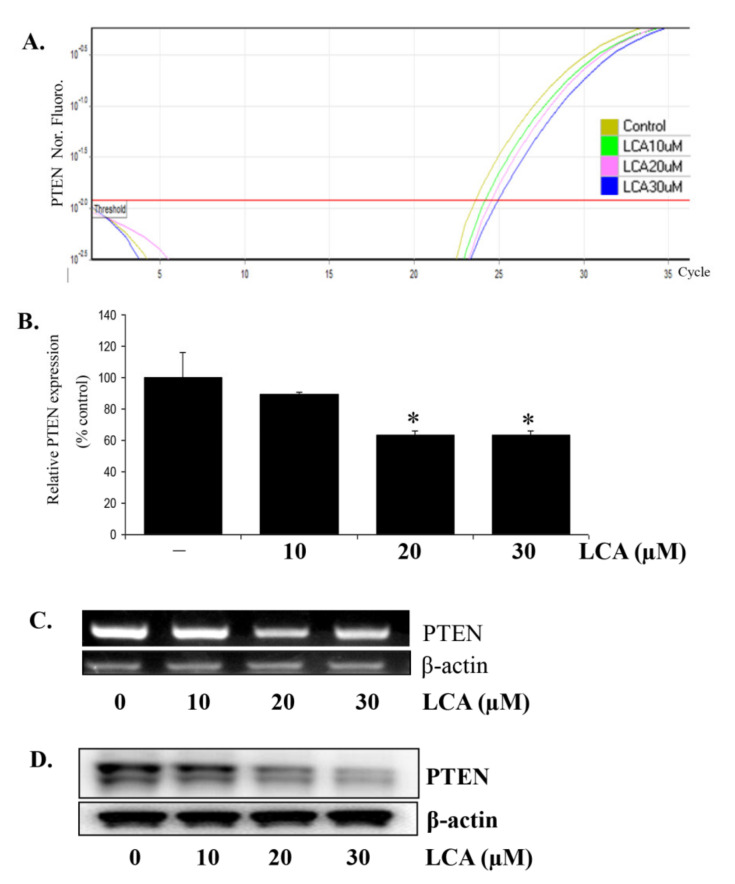
LCA inhibits PTEN expression in HCT116 cells. HCT116 cells were treated with LCA at 10 µM–30 µM concentration for 24 h. The cells were then extracted for total RNA and total protein for checking PTEN expression by Realtime-PCR (**A**,**B**), RT-PCR (**C**), and Western blot (**D**). * *p* < 0.05 versus control. The above data represent the means ± SD from triplicate measurements.

**Figure 6 ijms-22-10209-f006:**
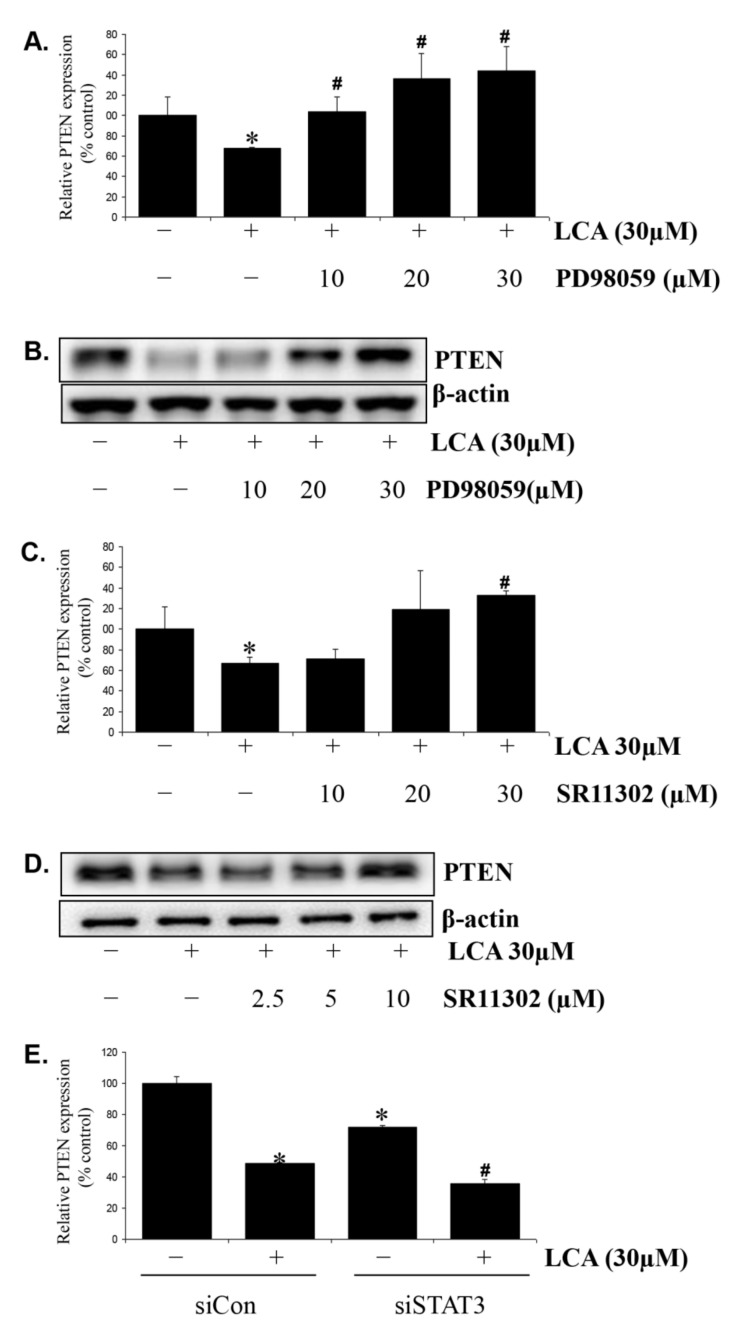
LCA inhibits PTEN expression via signals involved in the LCA-induced miR21 mechanism, Erk1/2-AP-1 and STAT3. HCT116 cells were treated with PD98059 at 10 µM–30 µM for 1 h prior to 30 µM LCA treatment for 24 h. The cells were then harvested and checked for PTEN expression by Realtime-PCR (**A**) and Western blot (**B**). HCT116 cells were treated to SR11302 at 10 µM–30 µM concentration for 1 h prior to LCA treatment and then subjected to check PTEN expression by Realtime-PCR (**C**) and Western blot (**D**). (**E**) HCT116 cells were transfected to 50 nM control siRNA (siCon) or siSTAT3, and then treated with 30 µM LCA for 24 h. The cells were further extracted for total RNA and checked for PTEN expression by Realtime-PCR. * *p* < 0.05 versus control; # *p* < 0.05 versus only LCA. The above data represent the means ± SD from triplicate measurements.

**Figure 7 ijms-22-10209-f007:**
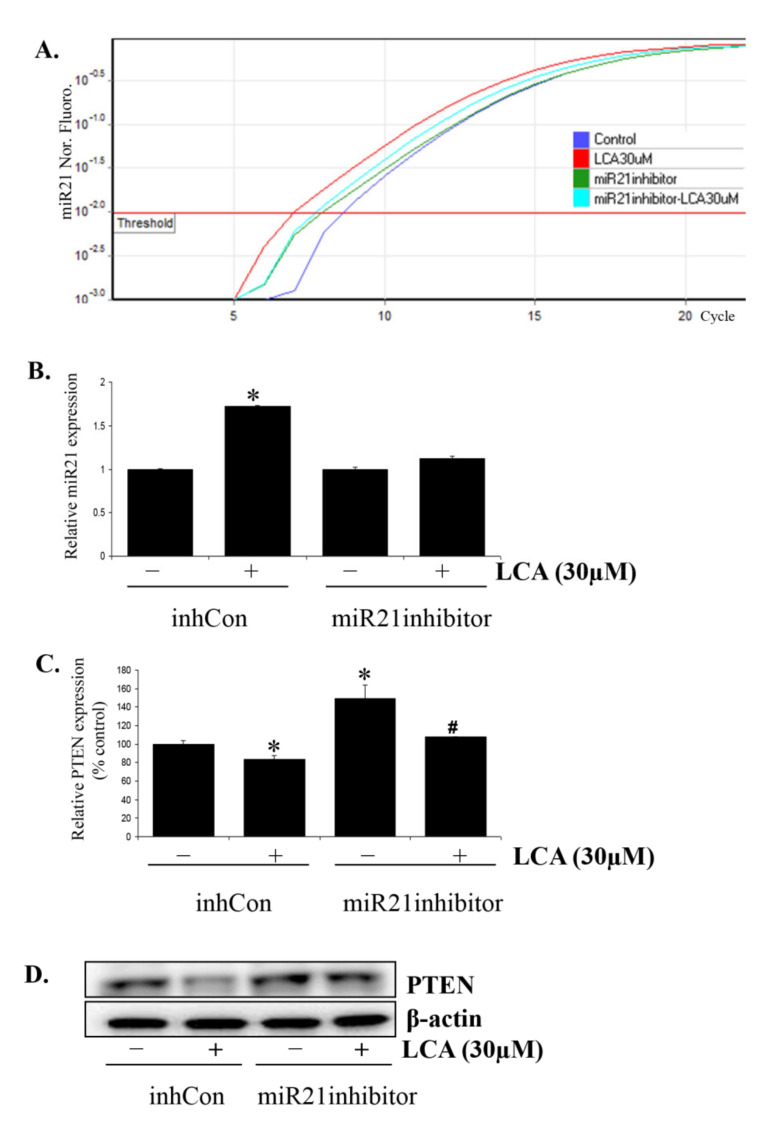
miR21-inhibitor rescues PTEN expression inhibited by LCA. HCT116 cells were transfected with 50 nM inhibitor negative control (inhCon) or miR21 inhibitor and treated with LCA for 24 h. The cells were then checked for miR21 expression by Realtime-PCR (**A**,**B**) and PTEN expression by Realtime-PCR (**C**) and Western blot (**D**). * *p* < 0.05 versus control; # *p* < 0.05 versus only LCA. The above data represent the means ± SD from triplicate measurements.

**Figure 8 ijms-22-10209-f008:**
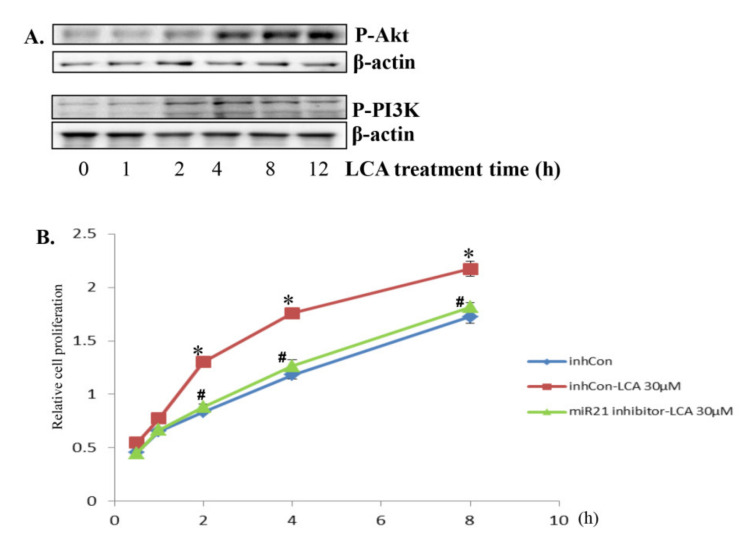
LCA activates Akt/PI3k signaling and enhances HCT116 cell proliferation. (**A**) 30 µM LCA was treated with HCT116 cells for 1 h–12 h and extracted for total protein for checking Phosphorylated Akt and PI3K by Western blot. (**B**) HCT116 cells transfected with 50 nM miR21 inhibitor or inhibitor negative control (inhCon) were treated with 30 µM LCA for 24 h. The cells were then checked for proliferation by EZ-Cytox assay. * *p* < 0.05 versus control; # *p* < 0.05 versus only LCA. The above data represent the means ± SD from triplicate measurements.

## Data Availability

The datasets used and/or analyzed during the current study are available from the corresponding author on reasonable request.

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
