# Peer review of "Lithocholic Acid Induces miR21, Promoting PTEN Inhibition via STAT3 and ERK-1/2 Signaling in Colorectal Cancer Cells"

_ijms, 2021, doi:10.3390/ijms221910209_

Round 1

Reviewer 1 Report

The paper is very interesting and well documented.

The introduction provide suficient information to introduce the reader to the topic of the research. The methods and the results are clearly presented.

Major points :

The discussions could be improved:

The paper suggest it is the first experimental work to investigate the relation between biliary acids and miR21 on CRCs. It would be better to rephase:..LCA instead of biliary acids, as there are some previous published articles regarding other biliary acids

The modulatory role of gut microbiota on the relation biliary scids- miR21 should be discussed

A short paragraph to summarise the finding of the research and possible clinical implications would be useful for the reader. 

Author Response

Dear Reviewer 

Thanks so much for your comments for our manuscript. We have carefully reviewed your comments and have addressed these comments as listed below.

Point 1: The paper suggest it is the first experimental work to investigate the relation between biliary acids and miR21 on CRCs. It would be better to rephase:..LCA instead of biliary acids, as there are some previous published articles regarding other biliary acids

Response to 1: We substituted “bile acid” by “LCA” in the following sentence:

“This study is the first to report that LCA affect miR21 expression in CRC cells, providing us with a better understanding of the cancer-promoting mechanism of bile acids that have been described as the very first promoters of CRC progression.” (Abstract, Page 1)

Point 2: The modulatory role of gut microbiota on the relation biliary scids- miR21 should be discussed

Response to 2: We supplement the discussion of the role of gut microbiota on the relation biliary acid-miR21 in the discussion part (Page 10)

Point 3: A short paragraph to summarise the finding of the research and possible clinical implications would be useful for the reader. 

Response to 3: A short paragraph was inserted at the end of the discussion part to summarize all findings of the study (page 11)

Sincerely,

Young Do Jung, M.D., Ph.D.

Department of Biochemistry

Chonnam National University Medical School

Hakdong 5, Kwangju, Korea 501-190

Phone: 82-62-220-4105, FAX: 82-62-223-8321

E-mail:[email protected]

Reviewer 2 Report

This study shows that lithocholic acid (LCA, 20-30uM) can induce the expression of oncogenic miR21, by STAT3 inhibition and Erk1/2 -> AP1 activation. Upregulated miR21 reduced the expression of PTEN and could sustain HCT116 CRC cell line proliferation, possibly via Akt/PI3K signaling.

The study is interesting and acts as a valuable bridge, merging and giving sense to previous observations on miR21 biology. As I report in the following comments, the main bias of the study is the use of a single cell line, chosen as the most responsive to LCA among those tested. This could be a strong limit to the value of the described observations, when applied to real-life CRC. I hope the authors could satisfy my suggestions to improve their valuable paper.

Comments

Most figures show microscopic fonts and have low definition. They appear blurry when magnified, thus they should be substituted by high-definition ones.

2.1         Only one CRC cell line (HCT116), showing multiple mutations on the pathways affected by miR21 (wnt, MAPK, PIK3K…) was used in this paper. Other cell lines tested (SW480, HT15 [HCT15?], DLD1, HT29) were reported to show lower miR21 induction, but data are completely missing. The use of the only cell line showing a strong induction of miR21 by LCA is a great bias for the study, as it suggests a cell-line specific activity, strongly reducing the possible involvement of the described biological effects in “real-life” colorectal cancer. All data should be repeated with at least two additional CRC lines. I would suggest to use NCI-H747, SKCO-1 and/or SNU-C1, all available on ATCC, as these lines carry few mutations, never affecting the PIK3K pathway (see http://cancerres.aacrjournals.org/content/suppl/2014/04/22/0008-5472.CAN-14-0013.DC1/Tab3.xls).

2.2         ΔAP1-1 seems to be the principal mediator of LCA activity, it would be very interesting to test if the same is observed using DCA. SR 11302, AP1 inhibitor, is usually used in the 1-2uM range, why was it tested in the 10-30uM range? Lower doses should be used, as high concentrations can cause off-target inhibitions without AP1 involvement.

2.3         Please note that PD98059 is a MEK1-2 inhibitor, blocking Erk1-2 phosphorylation, not a direct inhibitor of Erk1-2. This inhibitor works only at high concentrations [MEK1 (IC50 of 2-7 µM), MEK2 (IC50 of 50 µM)] and is not enough specific, as it also binds aryl hydrocarbon receptor and inhibits autophagy. I suggest to repeat the test using the highly specific PD0325901 [IC50 0.33 nM]. This additional test will confirm and strength the result. In this experimental setting, should also be included the detection an unrelated miR (not affected by LCA), as 24h is a very long time of incubation for cancer cells that rely on Erk1-2 activity for their survival, and a general and broad downregulation of gene expression is expected. Please test also the activity of the inhibitor in the absence of LCA. Parallel, western blot test of p-Erk1-2 levels should be performed in parallel, to demonstrate the modulation of the biochemical pathway. According to the methods, K97M transfection was not paralleled by a mock construct in controls, the same bias is apparently applied to siSTAT3 transfection (next paragraph), thus these tests are not completely reliable.

In some figures (2 b; S3 a-b; S4 a) each condition was apparently compared to its control, while all data should be normalized against the first control to show if transfectants have a different basal expression of the target.

2.4         HCT116 are ERBB3-mutated, thus a constitutive activation of STAT3 could be expected, this fact could explain why the mutSTAT3-BS1 construct increases also the basal induction of miR21.

2.5         (line 215-216) “In addition, siSTAT3, a STAT3 inhibitor, synergistically supported the inhibitory effect…” a silencing construct cannot be defined an inhibitor, please correct the sentence. The same apply to line 233 “miR21 inhibitor”. Fig 6b how do the authors explain the increased PTEN protein concentration observed only under the LCA+PD98059 (30uM) condition? RNA and protein data do not show the same dose-dependent modulation, moreover, as discussed above, PD98059 is not specific enough. Even in this case the WB detection of p-Erk1-2 phosphorylation status would be of great help. Fig 7d the difference in PTEN protein levels, in miR21 silenced cells w-w/o LCA, is barely detectable by eyes. Please, provide densitometric quantification of WB normalized against beta actin.

2.6         As reported in comments to 2.1 paragraph HCT116 carry PIK3CA mutation, thus the concomitant downregulation of PTEN can cause an amplification of Akt signaling. The authors should repeat these tests in at least two additional CRC cell lines with wild type PIK3CA, to show if the involvement of Akt, and the increased growth are equally strong under LCA treatment. According to the data in Fig 8, no statement could be done about the linkage of Akt activation and HCT116 cell growth. Indeed, also Erk1-2 are strong triggers of proliferation and are activated by LCA. The authors could mitigate the text, or perform additional experiments with specific inhibitors of these pathways. Also in this case, a mock-transfected control is mandatory to be compared to miR21-silencer transfected cells, as transfection could activate the interferon response, able to reduce cell proliferation.

Minor:

Please provide in full size all western blots as supplementary files.

(line 337) HT15 CRC cell line does not exist in ATCC collection, are they HCT15?

Author Response

Dear Reviewer

Thanks so much for your comments to our manuscript. We have carefully reviewed the comments provided by the two reviews and have addressed these comments as listed below.

Point 1: Most figures show microscopic fonts and have low definition. They appear blurry when magnified, thus they should be substituted by high-definition ones.

Response to 1: As the IJMS format, the figures for Peer Review were directedly inserted within the text, so it resolution is very low. We will directly send the figures in TIFF file with high resolution (600dpi) to the Journal for further steps.

Point 2: Only one CRC cell line (HCT116), showing multiple mutations on the pathways affected by miR21 (wnt, MAPK, PIK3K…) was used in this paper. Other cell lines tested (SW480, HT15 [HCT15?], DLD1, HT29) were reported to show lower miR21 induction, but data are completely missing. The use of the only cell line showing a strong induction of miR21 by LCA is a great bias for the study, as it suggests a cell-line specific activity, strongly reducing the possible involvement of the described biological effects in “real-life” colorectal cancer. All data should be repeated with at least two additional CRC lines. I would suggest to use NCI-H747, SKCO-1 and/or SNU-C1, all available on ATCC, as these lines carry few mutations, never affecting the PIK3K pathway (see http://cancerres.aacrjournals.org/content/suppl/2014/04/22/0008-5472.CAN-14-0013.DC1/Tab3.xls).

Response to 2: Missing the result of LCA-induced miR21 on other CRC cells was our mistake in the previous submission. For revised manuscript, we provided this result as Figure S1.A. Together, we also supplemented the result of  LCA-inhibited PTEN in different CRC cells as Figure 5S.

Point 3. ΔAP1-1 seems to be the principal mediator of LCA activity, it would be very interesting to test if the same is observed using DCA.

Response to 3: In this study, we just want to report the story of LCA-induced miR21. For DCA, For DCA, we briefly discussed in the  discussion session of the revised (Page 10-11).

Point 4: SR11302, AP1 inhibitor, is usually used in the 1-2uM range, why was it tested in the 10-30uM range? Lower doses should be used, as high concentrations can cause off-target inhibitions without AP1 involvement.

Response to 4: We performed the experiment with SR11302 at low concentration 1uM-3uM, 2.5uM-10uM, but only at 10uM-30uM we can get the expected result. For confirmation, we performed AP-1 promoter activity assay to determine SR11302 concentration range that could block LCA-induced AP-1. It showed that only 10-30uM SR11302 range could make the dose-dependent inhibition activity against  AP-1 activated by LCA. So, we think that this could because of strong AP-1 stimulatory effect of bile acid, we had to apply high concentration of AP-1 inhibitor.

Point 5:  Please note that PD98059 is a MEK1-2 inhibitor, blocking Erk1-2 phosphorylation, not a direct inhibitor of Erk1-2. This inhibitor works only at high concentrations [MEK1 (IC50 of 2-7 µM), MEK2 (IC50 of 50 µM)] and is not enough specific, as it also binds aryl hydrocarbon receptor and inhibits autophagy. I suggest to repeat the test using the highly specific PD0325901 [IC50 0.33 nM]. This additional test will confirm and strength the result. In this experimental setting, should also be included the detection an unrelated miR (not affected by LCA), as 24h is a very long time of incubation for cancer cells that rely on Erk1-2 activity for their survival, and a general and broad downregulation of gene expression is expected. Please test also the activity of the inhibitor in the absence of LCA.

Response to 5:  As the reviewer mentioned, high concentration of PD98059 could cause unexpected effect on cells. To avoid that, we supplemented the result of miR21 promoter activity assay using K97M, a dominant mutant MEK1/2 to block ERK1/2 activated by LCA as Figure S3.A. Both the data showed consistence results, so it is reliable to conclude the role of ERK1/2 in LCA-induced miR21 (S3.A).

Point 6: Parallel, western blot test of p-Erk1-2 levels should be performed in parallel, to demonstrate the modulation of the biochemical pathway.

Response to 6: The reviewer is correct  and p-Erk1/2 level stimulated by LCA was reported in our previous publish.

“Nguyen et al. Lithocholic Acid Stimulates IL-8 Expression in Human Colorectal Cancer Cells Via Activation of Erk1/2 MAPK and Suppression of STAT3 Activity. J Cell Biochem 2017, 118, (9), 2958-2967.”

In the manuscript, we also mentioned this referee to demonstrate the modulation of Erk1/2.

Point 7: According to the methods, K97M transfection was not paralleled by a mock construct in controls, the same bias is apparently applied to siSTAT3 transfection (next paragraph), thus these tests are not completely reliable.

Response to 7: All our transfection experiments were always performed with two parallel cells, including the cells transfected with the target and the cells transfected with mock construct control, as following: basic pSV2neo plasmid, an expressing vector of MEK1/2-K97M  as mock construct control of K97M mutant.; SignalSilence® Control siRNA (#6568, Cell Signaling) as mock contruct control of siSTAT3, inhibitor negative control (Cat. No. MIH000000, Applied Biological Materials Inc) as mock construct control of miR21 inhibitor. The method description (Page 13, 14) and related figures (4B, 6E, 7B-D, S3A-B) as well as the figure legends were edited to clear this information.

Point 8: In some figures (2 b; S3 a-b; S4 a) each condition was apparently compared to its control, while all data should be normalized against the first control to show if transfectants have a different basal expression of the target.

Response to 8: All data in the mentioned figures as well as all figures of the manuscript were normalized agaisnt the first control (the cells in normal culture condition or construct control transfected cells, without LCA treatment), not the specific condition control to show if  transfectants have a different basal expression of the target compared to normal control, as the reviewer mentioned.

Point 9: HCT116 are ERBB3-mutated, thus a constitutive activation of STAT3 could be expected, this fact could explain why the mutSTAT3-BS1 construct increases also the basal induction of miR21.

Response to 9: Human epidermal growth factor receptor 3 (ErbB3) belongs to the ErbB family of receptor tyrosine kinases (RTK), plays a significant role in cancer proliferation. Jak/Stat3 is one of downstream of ERBB3 signal.  There could be an interesting linking among ERBB3 mutation, STAT3 activation and miR21 expression in HCT116 CRC cells as the reviewer mentioned above. However, our result compared miR21 promoter activity of different promoter constructs in the same HCT116 cells. Logically, we can concluded that the basal induction of miR21 promoter activity of the mutSTAT3-BS1 construct compared to the naive construct is from STAT3-BS1 mutantion. This support our determination of STAT3 signaling in our interested mechanism, LCA-induced miR21.

Point 10: (line 215-216) “In addition, siSTAT3, a STAT3 inhibitor, synergistically supported the inhibitory effect…” a silencing construct cannot be defined an inhibitor, please correct the sentence. The same apply to line 233 “miR21 inhibitor”.

Response to 10: We corrected it as STAT3 silencer. miR21 inhibitor is official name from Provider (Applied Biological Materials Inc., Richmond, BC, Canada) so we follow it.)

Point 11: Fig 6b how do the authors explain the increased PTEN protein concentration observed only under the LCA+PD98059 (30uM) condition? RNA and protein data do not show the same dose-dependent modulation, moreover, as discussed above, PD98059 is not specific enough. Even in this case the WB detection of p-Erk1-2 phosphorylation status would be of great help.

Response to 11: We changed another image of this result to show dose-dependent inhibition activity of PD98059 against LCA-inhibited PTEN. We also supplemented the densitometric quantification of  PTEN expression normalized against beta actin of this WB figure as Figure S5.A to support it.

Point 12: Fig 7d the difference in PTEN protein levels, in miR21 silenced cells w-w/o LCA, is barely detectable by eyes. Please, provide densitometric quantification of WB normalized against beta actin.

Response to 12: The densitometric quantification of PTEN expression normalized against beta actin  of this figure was supplemented as Figure S5.B. The densitometric analysis was supplemented in the method part (Page 14).

Point 13: As reported in comments to 2.1 paragraph HCT116 carry PIK3CA mutation, thus the concomitant downregulation of PTEN can cause an amplification of Akt signaling. The authors should repeat these tests in at least two additional CRC cell lines with wild type PIK3CA, to show if the involvement of Akt, and the increased growth are equally strong under LCA treatment. According to the data in Fig 8, no statement could be done about the linkage of Akt activation and HCT116 cell growth. Indeed, also Erk1-2 are strong triggers of proliferation and are activated by LCA. The authors could mitigate the text, or perform additional experiments with specific inhibitors of these pathways.

Response to 13: As the reviewer’s suggestion, actually we have performed experiments using Erk1/2, PI3k , and Akt inhibitor to track their role in LCA-induced cell proliferation but we can not get clear results to go to a final conclusion. So in this part, we just want to show the activation of  PI3k/Akt  pathway under LCA treatment, as one of downstream of PTEN inhibition.

In this study, we just confirmed the role of LCA-induced miR21 mediated for cell proliferation stimulation, but have not figured out the relationship among the cell proliferation acceleration with PI3k/Akt or Erk1/2 pathways yet. It should be left as an open question needing further proven. The statement (Page 9) and discussion (Page 11) parts of this result was edited for fixing with the obtained result.

Point 14: Also in this case, a mock-transfected control is mandatory to be compared to miR21-silencer transfected cells, as transfection could activate the interferon response, able to reduce cell proliferation.

Response to 14: As mentioned above, the inhibitor negative control (Cat. No. MIH000000, Applied Biological Materials Inc) was used as mock contruct control of miR21 inhibitor. So in this transfection experiment to access cell proliferation, we also follow the similar way. The cells were transfected with miR21 inhibitor or its mock contruct control, the inhibitor negative control, then treated with LCA to access cell proliferation.The experiment description (Page 14) and Figure 8B legend were edited to clear it.

 Point 15: Please provide in full size all western blots as supplementary files.

Response to 15: Full size all western blots and agarose gels were supplemented as “Original images” file.

Point 16: (line 337) HT15 CRC cell line does not exist in ATCC collection, are they HCT15?

Response to 16: It exactly HCT15. We corrected it.

Should any questions raise, please feel free to contact me.

Sincerely,

Young Do Jung, M.D., Ph.D.

Department of Biochemistry

Chonnam National University Medical School

Hakdong 5, Kwangju, Korea 501-190

Phone: 82-62-220-4105, FAX: 82-62-223-8321

E-mail:[email protected]

Round 2

Reviewer 1 Report

The manuscript have been revised according to the comments. I recommend accept for publishing

Author Response

September 15th, 2021

Dear Reviewer

RE: ijms-1330295

Title: LCA induces miR21, promoting PTEN inhibition via STAT3 and ERK-1/2

           Thanks so much for all your kind comments for our manuscript. For final revision, we carefully checked again language and grammar of the whole manuscript and made some edition in the text to make it better. All edition was marked up using the “track changes” function.

           Should any questions raise, please feel free to contact me.

Sincerely,

Young Do Jung, M.D., Ph.D.

Department of Biochemistry

Chonnam National University Medical School

Hakdong 5, Kwangju, Korea 501-190

Phone: 82-62-220-4105, FAX: 82-62-223-8321

E-mail:[email protected]

Reviewer 2 Report

I thank the authors for their answers to my perplexities.

In the future I suggest them to perform all the experiments in at least 3 different cell lines (with different driving mutations), as this is a requested standard for many journals. Though, I will not insist on this point for the present study. Also, the use of the best inhibitors available (working at very low dilutions, without demonstrated off-target activities) should be a quality starting-point for biochemical pathway studies.

As a final minor revision, please correct the title of Suppl fig. S5 (also in the main text, line 503) as it states that the results are shown for the HCT116 cell line only, while four cell lines are shown.

Best regards

Author Response

September 15th, 2021

Dear Editor

RE: ijms-1330295

Title: LCA induces miR21, promoting PTEN inhibition via STAT3 and ERK-1/2

        Thanks so much for all your kind comments for our manuscript. They are really helpful advices for not only this our manuscript but also our further studies.

         We have carefully reviewed and addressed your minor suggestion as listed below.

  1. We corrected the title of Fig. S5 in supplementary file as well as in main text.
  2. We carefully checked again language and grammar of the whole manuscript and made some edition in the text to make it better. All edition was marked up using the “track changes” function.

         Should any questions raise, please feel free to contact me.

Sincerely,

Young Do Jung, M.D., Ph.D.

Department of Biochemistry

Chonnam National University Medical School

Hakdong 5, Kwangju, Korea 501-190

Phone: 82-62-220-4105, FAX: 82-62-223-8321

E-mail:[email protected]